# Expansins in Salt and Drought Stress Adaptation: From Genome-Wide Identification to Functional Characterisation in Crops

**DOI:** 10.3390/plants14091327

**Published:** 2025-04-28

**Authors:** Siarhei A. Dabravolski, Stanislav V. Isayenkov

**Affiliations:** 1Department of Biotechnology Engineering, Braude Academic College of Engineering, Snunit 51, Karmiel 2161002, Israel; sergedobrowolski@gmail.com; 2Institute of Agricultural and Nutritional Sciences, Martin Luther University Halle-Wittenberg, Betty-Heimann-Strasse 3, 06120 Halle, Germany

**Keywords:** expansin, cell-wall loosening, salt stress, drought stress, abiotic stress, stress tolerance

## Abstract

Expansins are cell wall-modifying proteins that play a pivotal role in plant growth, development, and stress adaptation to abiotic stress. This manuscript explores the functions of expansins in salt and drought stress responses across multiple plant species, highlighting their involvement in cell wall loosening, transcriptional regulation, ion and osmotic homeostasis, and phytohormone signalling. Genome-wide identification and expression analyses revealed differential regulation of expansin genes under abiotic stress conditions. In *Nicotiana tabacum*, overexpression of *NtEXPA4* and *NtEXPA11* promoted root elongation and ion homeostasis, improving salt and drought tolerance. Similarly, *Brassica rapa BrEXLB1* was found to modulate root architecture and phytohormone-mediated stress responses. In *Oryza sativa*, *OsEXPA7* was linked to cation exchange and auxin signalling under salt stress conditions. Conversely, in *Populus trichocarpa*, *PtEXPA6* exhibited a negative regulatory role in salt stress tolerance, highlighting species-specific differences in expansin function. Expansins also contribute to reactive oxygen species (ROS) homeostasis, as observed in transgenic plants with increased activities of SOD, POD, APX, and CAT, which reduced oxidative damage under stress. Additionally, enhanced accumulation of soluble sugars and proline in expansin-overexpressing plants suggests their involvement in osmotic adjustment mechanisms. The interplay between expansins and ABA, auxins, and ethylene further underscores their role in integrating mechanical and hormonal stress responses. Despite substantial progress, limitations remain in understanding the broader regulatory networks influenced by expansins. Future research should focus on elucidating their downstream molecular targets, transcriptional interactions, and functional diversity across different plant species. Expansins represent promising candidates for improving crop resilience to environmental stress, making them valuable targets for future breeding and biotechnological approaches.

## 1. Introduction

Abiotic stresses such as salinity and drought are among the most significant factors limiting plant growth, development, and crop productivity worldwide. As global climate change accelerates, these stresses are becoming increasingly frequent and severe, underscoring the urgent need to develop crops with enhanced stress resilience [1]. Plants have evolved intricate mechanisms to perceive and respond to environmental challenges, involving complex physiological, biochemical, and molecular adaptations [2]. A key aspect of these adaptive responses involves the modification of cell wall properties, which directly influences plant growth and stress tolerance [3].

Expansins are a family of cell wall-loosening proteins that play a central role in modulating cell wall extensibility without hydrolysing cell wall polymers [4]. Originally identified for their role in cell expansion and growth, expansins are now recognised as critical regulators of various developmental processes and responses to abiotic stresses. By modifying the mechanical properties of the cell wall, expansins facilitate root elongation, stomatal regulation, and tissue plasticity, enabling plants to maintain growth and water balance under adverse conditions [5]. In addition to their mechanical functions, emerging evidence suggests that expansins interact with diverse molecular networks, including transcriptional regulators, antioxidant defence systems, and phytohormone signalling pathways, contributing to a coordinated stress response [6].

Genome-wide analyses have identified numerous expansin genes differentially expressed under salt and drought stress across multiple plant species, highlighting both conserved and species-specific roles [6]. Functional studies involving overexpression or knockdown of expansin genes have demonstrated their ability to enhance ion homeostasis, osmotic adjustment, and reactive oxygen species (ROS) scavenging, thereby improving plant tolerance to abiotic stress [5]. Nevertheless, the regulatory networks and downstream targets of expansins remain incompletely understood, and their practical application for crop improvement is still in its early stages [7].

This manuscript provides a comprehensive analysis of the roles of expansins in salt and drought stress responses, drawing on findings from multiple plant species, including *Nicotiana tabacum*, *Brassica rapa*, *Oryza sativa*, and *Populus trichocarpa*. Particular emphasis is placed on their functions in cell wall remodelling, antioxidant defence, osmoprotectant accumulation, and hormonal crosstalk. By synthesising current knowledge and highlighting key gaps, this work aims to provide insights into the potential of expansins as targets for enhancing crop resilience under challenging environmental conditions.

## 2. Expansins: Structure and Function in Plant Growth

Expansins are extracellular proteins essential for cell wall (CW) loosening, relaxation, and extension during pH-dependent ‘acid growth’. They are found in all plants, though gene loss has been observed in aquatic species [4,5,8]. Expansins also occur in fungi and bacteria, likely due to horizontal gene transfer. Their presence in cellulose-utilising eukaryotic micro-organisms suggests an evolutionary origin in ancient marine microbes predating land plants [9,10,11]. In bacteria and fungi, expansins facilitate plant–microbe interactions and are used industrially for lignocellulose degradation in biofuel production [12,13,14,15,16,17].

Expansins in vascular plants are classified into four subfamilies: α-expansin (EXPA), β-expansin (EXPB), expansin-like A (EXLA), and expansin-like B (EXLB) [7,18]. EXPA and EXPB mediate CW loosening, with *EXPA* forming the largest subfamily, comprising 26 genes in *Arabidopsis thaliana*, compared to *6 EXPB*, *3 EXLA*, and *1 EXLB* [19]. Structurally, expansins are torpedo-shaped proteins with two domains (D1 and D2) connected by a linker. D1, related to GH45 enzymes, lacks catalytic activity, while D2 resembles CBM family 63. Both domains are crucial for CW loosening [6,20,21]. Computational models, structural analyses, and studies of related proteins have provided key insights into expansin function [22,23,24,25].

Bacterial expansins, such as *Bacillus subtilis* EXLX1, bind crystalline cellulose through conserved aromatic residues in D2, disrupting hydrogen bonds and increasing substrate accessibility [26,27]. They enhance cellulase activity in lignocellulose hydrolysis, benefiting industrial applications [28,29,30]. Adsorption studies indicate that EXLX1 activity is enhanced at low cellobiose and xylose concentrations but inhibited at high concentrations [31,32,33]. Notably, although bacterial expansins exhibit stronger binding than plant α-expansins, mutant variants fail to induce CW creep [34].

According to the loosening theory, well-hydrated, non-growing cells maintain osmotic equilibrium, with wall stresses counterbalancing turgor pressure. In actively growing cells, CW loosening—primarily mediated by expansins—reduces tensile stress and turgor pressure, prompting water influx. This expansion restores turgor and wall stress, driving cell growth. Crucially, growth is initiated by CW loosening and subsequent turgor changes, rather than the reverse [6,35,36].

The ‘acid growth theory’ [37] proposes that auxin stimulates proton (H^+^) extrusion into the apoplast, activating expansins and enabling CW loosening and cell expansion [8,38]. This process is regulated by plasma membrane-localised P-type H^+^-ATPases, which acidify the apoplast by pumping protons into the CW matrix [39,40,41,42]. The TIR1/AFB-Aux/IAA auxin signalling pathway controls this activity by upregulating *SMALL AUXIN UP-RNA 19* (*SAUR19*) expression [43,44]. *SAUR19* inhibits *TYPE 2C PROTEIN PHOSPHATASES* (*PP2C*), keeping the H^+^-ATPase in an active state [45,46]. Proton extrusion also hyperpolarises the plasma membrane, activating K^+^ channels to maintain intracellular osmotic potential and ensure sustained water uptake and turgor for CW expansion [47,48].

Cytokinins upregulate *H^+^-ATPase* genes (e.g., *AHA2* and *AHA7*), facilitating EXPA1-mediated elongation in root transition zones [49,50,51]. Brassinosteroids similarly regulate cell elongation by modulating H^+^-ATPase activity, with K^+^ antiport via CNGC10 balancing H^+^ efflux and maintaining membrane potential. In roots, a pH gradient—marked by H^+^ influx in the meristematic zone and H^+^ efflux in the transition zone—may regulate elongation [52,53,54]. Thus, spatially controlled expansin activity and precise H^+^ flux regulation are essential for CW acidification and cell expansion.

Drought is a major abiotic stress that threatens global food security by impairing plant growth and productivity [55,56,57]. It induces osmotic stress, disrupting cell division, halting growth and development, and increasing ROS production, ultimately reducing yield quality and quantity [58,59,60]. For example, expansins expression plays a crucial role in the ability of the well-known “resurrection plant” *Craterostigma plantagineum* Hochst. to survive desiccation [61]. Salinity, another severe stressor, combines osmotic and ionic effects, disrupting Na^+^ and Cl^–^ homeostasis, impairing water and nutrient uptake, damaging cell membranes, and reducing photosynthesis, leading to lower crop yields [62,63,64]. Advances in omics technologies and gene editing have recently provided insights into the molecular mechanisms underpinning expansin-mediated improvements in plant drought and salinity tolerance. Further, in this manuscript, we discuss the roles of expansins in salt and drought stress responses across various plant species, focusing on their involvement in cell wall remodelling, transcriptional regulation, ion and osmotic homeostasis, and phytohormone signalling. We examine genome-wide expression patterns, functional analyses, and transgenic studies that reveal how specific expansin genes, such as *NtEXPA4* and *NtEXPA11* in *Nicotiana tabacum*, *BrEXLB1* in *Brassica rapa*, and *OsEXPA7* in *Oryza sativa*, contribute to stress adaptation. Additionally, we explore the connections between expansins and ROS scavenging systems, osmoprotectant accumulation, and hormonal crosstalk, providing a comprehensive overview of their regulatory functions. Finally, we highlight key limitations in current research and propose future directions for enhancing our understanding of expansin-mediated stress tolerance.

## 3. Genome-Wide Identification and Expression Analysis Under Stress Conditions in Model and Crop Species

Despite extensive studies on expansin function in cell wall modification, their role in plant responses to environmental stress remains an active area of research. To better understand their involvement, genome-wide identification and expression profiling under stress conditions provide valuable insights. Thus, in tobacco (*Nicotiana tabacum* L.), *NtEXPA1*, *NtEXPA4*, and *NtEXPA5* were predominantly expressed in the shoot apices and young leaves, whereas *NtEXPA6* accumulated at high levels in calluses rich in undifferentiated cells. The expression of *NtEXPA1*, *NtEXPA4*, and *NtEXPA5* in young leaves was upregulated by cytokinins, auxins, and gibberellins. Additionally, exposure to salt, drought, heat, cold, and abscisic acid (ABA) significantly increased the transcription of *NtEXPA1*, *NtEXPA4*, and *NtEXPA5*, while these stresses did not affect *NtEXPA6* expression [65] (see Table 1).

Similar to tobacco, where expansin expression was influenced by both hormones and abiotic stress, potato (*Solanum tuberosum* L.) expansins also exhibited varying expression patterns across different tissues and under stress conditions (Table 1). In potato, *StEXPB2* was most abundantly expressed in young tubers, while *StEXPA11*, *StEXPA16*, *StEXPA4*, *StEXPA14*, and *StEXLA1* exhibited relatively high expression across most tested tissues. The response of potato expansins to hormonal treatments varied; *StEXPA7* and *StEXPA18* were upregulated by IAA, gibberellic acid (GA_3_), and BAP, whereas all *StEXLB* genes were induced by ABA and GA_3_. Under NaCl and mannitol treatments, *StEXPA8*, *StEXPA19*, and *StEXPB2* were upregulated, whereas *StEXPA4* and *StEXLB4* were downregulated. A total of 23 *StEXP* genes responded to water stress, and the expression of 18 genes changed under high-temperature stress. Notably, *StEXLB6* exhibited the highest expression levels under both drought and high-temperature stress, while *StEXPB2* showed the most significant downregulation in response to high-temperature stress. Interestingly, five genes (*StEXPA1*, *StEXPA21*, *StEXPA23*, *StEXPA24*, and *StEXPB5*) remained unaffected by abiotic stresses or hormonal treatments [66].

In a comparable manner, wheat (*Triticum aestivum* L.) expansins showed tissue-specific expression profiles, with notable differences in response to abiotic stresses, further highlighting the complexity of expansin regulation across species. Some wheat expansins exhibited organ-specific expression, while others showed broad expression patterns (Table 1). For instance, five genes (*TaEXPA4-A*, *TaEXPA5-A*, *TaEXPA6-A*, *TaEXPA8-A*, and *TaEXPB8-A*) were exclusively expressed in roots, four genes (*TaEXPA1-A*, *TaEXPA1-D*, *TaEXPA12-A*, and *TaEXPB1-A*) were highly expressed in leaves, and *TaEXPA2-D*, *TaEXPB1-B*, *TaEXPB2-A*, and *TaEXPB4-A* displayed high expression in seeds. Interestingly, 14 genes showed no detectable expression at any tested growth stage across five organs (grain, leaf, root, spike, and stem). In leaves, polyethylene glycol (PEG) treatment upregulated *TaEXPA3-A*, *TaEXPA9-A*, *TaEXPB2-A*, *TaEXPB4-A*, *TaEXPB7-A*, *TaEXPB9-A*, and *TaEXPB10-A*, while salt stress upregulated *TaEXPA3-A*, *TaEXPA9-A*, *TaEXPB2-A*, *TaEXPB4-A*, and *TaEXPB10-A*. The remaining expansins exhibited a general downregulation under salt stress. In roots, PEG treatment upregulated *TaEXPA3-A*, *TaEXPA4-A*, *TaEXPA5-A*, *TaEXPA6-A*, *TaEXPA7-A*, *TaEXPA8-A*, *TaEXPB1-A*, *TaEXPB1-B*, *TaEXPB7-A*, *TaEXPB8-A*, and *TaEXPB10-A*, while most other expansins were downregulated. Salt stress upregulated *TaEXPA3-A*, *TaEXPA5-A*, *TaEXPA6-A*, *TaEXPA7-A*, *TaEXPA8-A*, *TaEXPA12-A*, *TaEXPB2-A*, *TaEXPB4-A*, *TaEXPB7-A*, and *TaEXPB10-A*, with the remaining genes generally showing a downregulated expression pattern [67].

Likewise, in barley (*Hordeum vulgare* L.), the expression of expansins was predominantly observed in embryonic and root tissues, with specific responses to drought stress, suggesting potential roles in stress adaptation across different plant species (Table 1). In barley, most *HvEXPAs* were mainly expressed in embryonic and root tissues, whereas *HvEXPBs* and *HvEXLAs* exhibited diverse expression patterns across 16 tissues at different developmental stages. For instance, *HvEXPA3*, *HvEXPA4*, and *HvEXPA22* were specifically expressed in caryopses, while *HvEXPB5*, *HvEXPB6*, and *HvEXPB11* were exclusively detected in roots. In contrast, *HvEXPA13*, HvEXPA14, *HvEXPA23*, and *HvEXLA5* showed little to no expression in nearly all tissues. Under drought stress, *HvEXPB5* and *HvEXPB6* were significantly upregulated in roots, whereas *HvEXPA22*, *HvEXPB1*, *HvEXLA4*, *HvEXLA5*, and *HvEXLA6* remained undetectable under both normal and water-deficit conditions. The strong upregulation of *HvEXPB5* and *HvEXPB6* in response to drought suggests their potential involvement in drought tolerance in barley, warranting further experimental investigation [68].

This section highlights the genome-wide identification and expression analysis of expansins in response to various stress conditions across different plant species. The expression profiles of expansins in tobacco, potato, wheat, and barley revealed a complex and species-specific regulation, with distinct patterns in different tissues and under various stress treatments. Notably, expansins in tobacco and potato were found to be highly responsive to both hormonal signals and abiotic stresses such as salt, drought, and heat, with some genes exhibiting upregulation and others showing no response. Wheat expansins displayed an organ-specific expression, with several genes being induced by PEG and salt stress, while barley expansins demonstrated a clear involvement in drought tolerance, particularly in roots. These findings highlight the functional diversity of expansins and their potential role in plant stress adaptation. Further studies are needed to unravel the precise mechanisms by which expansins contribute to stress tolerance and their potential application in improving crop resilience to environmental challenges.

## 4. Functional Analysis of Expansins in Salt and Drought Stress Tolerance

While genome-wide studies highlight the potential role of expansins in stress responses, direct functional validation is necessary to determine their precise contributions. This can be achieved through genetic manipulation in both heterologous (Section 4.1) and native plant systems (Section 4.2).

### 4.1. Expansin Ectopic Expression in Heterologous Systems: Enhancing Stress Tolerance in Model and Crop Plants

The role of expansins in drought and salt tolerance has been extensively studied through ectopic expression in various model species. For instance, the constitutive expression of wheat *EXPA2* (*TaEXPA2*) in tobacco was used to investigate its role in plant drought tolerance and development. *TaEXPA2*-expressing tobacco plants exhibited improved seed production by increasing capsule numbers without altering plant growth patterns. Moreover, transgenic tobacco accumulated higher levels of proline and displayed an enhanced antioxidant status, as indicated by lower reactive oxygen species (ROS) and malondialdehyde (MDA) content, as well as reduced relative electrical conductivity (Figure 1 and Figure 2). These changes contributed to increased drought tolerance compared to wild-type plants [70].

Building on the findings from *TaEXPA2* in drought tolerance, further studies also demonstrated that *TaEXPA2* plays a significant role in enhancing salt stress tolerance in transgenic tobacco. Thus, *TaEXPA2* expression in tobacco enhanced salt stress tolerance, as evidenced by higher survival and germination rates, longer root length, and an increased number of lateral roots and green leaves under salt stress compared to wild-type plants. Additionally, transgenic plants exhibited higher chlorophyll content, increased accumulation of proline and soluble sugars, and lower Na^+^ but higher K^+^ concentrations than control plants. Antioxidant capacity was also improved, as indicated by increased activity and expression of *superoxide dismutase* (*SOD*), *catalase* (*CAT*), *ascorbate peroxidase* (*APX*), and *peroxidase* (*POD*), along with lower MDA and H_2_O_2_ content (Figure 1 and Figure 2). Furthermore, both ABA and NaCl treatment induced *TaEXPA2* expression and protein accumulation, whereas the application of an ABA inhibitor reversed these effects, suggesting the involvement of ABA signalling in *TaEXPA2*-mediated salt stress tolerance [71].

In a similar context, the expression of another expansin, *AtEXPA18* from *Arabidopsis thaliana*, in tobacco plants also resulted in improved growth under drought stress, indicating the broader potential of expansins in stress tolerance. Transgenic tobacco plants expressing *AtEXPA18* demonstrated improved growth parameters, including increased height, stem diameter, leaf area and number, and dry root weight, as well as elevated cytokinin and ABA levels in leaves under drought stress [72].

Expanding the investigation to other species, the expression of wheat *EXPA7-B (TaEXPA7-B*) in rice also yielded positive results, enhancing plant growth and stress resilience under salt stress. *TaEXPA7-B* expression in rice enhanced plant height, root length, and the number of lateral roots compared to wild-type plants (Figure 2). Under salt stress, transgenic plants accumulated higher levels of osmotic regulators (soluble sugars, soluble proteins, and proline) and cell wall-related substances (lignin, cellulose, and hemicellulose) while exhibiting increased antioxidant enzyme activity (SOD, POD, CAT) compared to wild-type plants [73].

In addition to wheat and rice, the expression of the *PttEXPA8* gene from *Populus tomentosa* in tobacco further exemplifies the role of expansins in enhancing abiotic stress resistance, showing improvements across a range of environmental stresses. Transgenic tobacco plants expressing the *PttEXPA8* gene exhibited improved resistance to a range of abiotic stresses, including heat, drought, salt, cold, and cadmium. Notably, transgenic lines demonstrated the highest improvements in seed germination and primary root length. In mature plants, *PttEXPA8*-expressing lines exhibited enhanced growth under drought, heat, and cold stress compared to wild-type plants. Additionally, transgenic leaves retained higher soluble sugar content, greater SOD activity, and lower relative electrolyte leakage under drought, heat, and salt stress than wild-type leaves. Furthermore, *PttEXPA8*-expressing plants exhibited lower MDA content under drought, heat, salt, cold, and Cd^2+^ stress compared to wild-type plants [74].

Additionally, in salt-sensitive and salt-tolerant willow (*Salix matsudana Koidz.*) plants, the expression of *SmEXPA23* highlighted its potential role in salt resistance, supporting the growing body of evidence on expansin involvement in stress tolerance mechanisms. Thus, a comparison of salt-tolerant and salt-sensitive willow plants subjected to salt stress revealed that *SmEXPA23* expression was 1.83 times higher in salt-tolerant plants compared to salt-sensitive ones, suggesting a role for *SmEXPA23* in salt resistance. Accordingly, tobacco plants expressing *SmEXPA23* exhibited reduced relative electrolyte leakage (REL) and MDA content under salt stress. Furthermore, transgenic tobacco plants accumulated significantly less Na^+^ in roots and leaves (47% and 64% lower, respectively) while taking up more K^+^ (increased by 29% and 33% in roots and leaves, respectively) compared to wild-type plants (Figure 1 and Figure 2) [75].

Similarly, *SmEXPA13* expression in tobacco plants resulted in enhanced salt tolerance, reinforcing the importance of expansins in regulating ion homeostasis and mitigating stress-related damage. Tobacco plants expressing *SmEXPA13* displayed a reduction in REL and MDA content (by 24.36% and 26.4%, respectively) and accumulated less Na^+^ in roots and leaves (by 58% and 63%, respectively) while increasing K^+^ uptake (by 78% and 47%, respectively) compared to wild-type plants (Figure 1) [76]. Collectively, these results indicate that expansin genes contribute to salt and drought tolerance by regulating osmotic regulators, morphogenesis-related processes, antioxidant enzyme activities, and phytohormone content.

### 4.2. Functional Characterisation of Expansin Mutants and Overexpression Lines in Native Species


*Nicotiana tabacum*


Transgenic tobacco plants overexpressing *NtEXPA1* and *NtEXPA5* exhibited enhanced tolerance to salt stress, as indicated by increased stem height and reduced water loss due to a lower stomatal density per unit leaf area compared to wild-type plants. In contrast, *NtEXPA4*-silenced plants displayed a salt-sensitive phenotype, characterised by reduced stem height and increased water loss under salt stress. These findings suggest that *NtEXPA1*, *NtEXPA4*, and *NtEXPA5* function as positive regulators of salt tolerance in tobacco [65].

In addition to the findings on salt tolerance, further analysis of *NtEXPA4* revealed its dual role in regulating both abiotic and biotic stress responses. Thus, *NtEXPA4* overexpression resulted in lower REL, higher fresh weight, increased accumulation of soluble sugars, proline, and ABA, and upregulated expression of several stress-responsive genes, including *P5CS* (proline biosynthesis), *SOS1* (plasma membrane Na^+^/H^+^ antiporter), and *ABA2* (ABA biosynthesis). Conversely, *NtEXPA4*-silenced lines exhibited hypersensitivity to salt and drought stress. Interestingly, while *NtEXPA4* overexpression conferred improved abiotic stress tolerance, it also increased susceptibility to the viral pathogen TMV-GFP and accelerated disease progression caused by *Pseudomonas syringae* DC3000 compared to wild-type and RNAi lines. These findings indicate that *NtEXPA4* is a key regulator of tobacco responses to both abiotic and biotic stresses [77].

Furthermore, overexpression of another tobacco expansin gene, *NtEXPA11*, exhibited a more pronounced effect on plant growth and stress resilience, particularly under salt and drought conditions. *NtEXPA11* overexpression promoted plant growth under normal conditions, leading to increased root density, root biomass, lateral root number, leaf area and number, plant height, internode length, and stem thickness. At the cellular level, transgenic plants displayed enlarged pith and parenchyma cells compared to wild-type plants. Additionally, *NtEXPA11* overexpression enhanced salt stress tolerance, as evidenced by larger leaves, a greater number of lateral roots, higher chlorophyll content, lower H_2_O_2_ accumulation, and a higher survival rate under salt stress. Under drought conditions, transgenic plants exhibited slower water loss and faster recovery upon re-watering compared to wild-type plants. These findings highlight the morphological and physiological alterations associated with *NtEXPA11* overexpression, which contribute to improved growth and tolerance to salt and drought stress [78].


*Brassica rapa*


Expanding the study to other species, the recent characterisation of *BrEXLB1* in *Brassica rapa* further illustrated the complex role of expansins in modulating stress responses, particularly in relation to phytohormone signalling. The characterisation of *BrEXLB1*, an expansin-like B1 gene in *Brassica rapa*, demonstrated its responsiveness to phytohormones. *BrEXLB1* expression was upregulated by indole-3-acetic acid (IAA), ABA, salicylic acid (SA), and ethylene, whereas cytokinin (CK), GA_3_, and jasmonic acid (JA) had no significant effect. Expression analysis under abiotic stress conditions revealed that drought, salt, osmotic, and oxidative stresses upregulated *BrEXLB1*, whereas cold stress and biotic stress (including *Turnip mosaic virus* infection, *Pectobacterium carotovorum*, and clubroot disease) downregulated its expression. Functional analysis of *BrEXLB1*-overexpressing lines revealed significantly reduced germination rates under both normal and drought conditions compared to wild-type plants. However, *BrEXLB1* overexpression was positively correlated with drought tolerance and improved photosynthetic efficiency during vegetative growth. Collectively, these findings suggest that *BrEXLB1* positively regulates drought tolerance and photosynthesis while negatively impacting seed germination [79].


*Populus trichocarpa*


In contrast to the effects observed in *Brassica rapa*, studies on expansin genes in *Populus trichocarpa* revealed a negative regulatory role for expansin A6 (*PtEXPA6*) in salt tolerance, adding another layer of complexity to the functional roles of expansins in stress resistance. Under normal conditions, *PtEXPA6* expression remained stable, whereas short-term salt stress induced its upregulation, followed by downregulation under prolonged stress. Transgenic *Populus alba* × *P. tremula* var. *glandulosa* (84K) and *Populus tremula* × *P. alba* INRA ‘717-1B4’ lines overexpressing *PtEXPA6* exhibited a greater reduction in stem height and diameter following long-term (15-day) salt stress compared to wild-type plants. Furthermore, transgenic lines accumulated higher Na^+^ concentrations in roots, stems, and leaves; exhibited a greater decline in net photosynthesis rate, and displayed lower antioxidant enzyme (SOD, POD, and CAT) expression and activity than wild-type plants. Notably, Na^+^ efflux from root xylem and leaf petiole vascular bundles was elevated in *PtEXPA6*-overexpressing plants, leading to excessive Na^+^ accumulation and reduced salt tolerance. Increased root contractility and extensibility in transgenic plants suggested that *PtEXPA6*-mediated cell wall loosening facilitated Na^+^ radial translocation into the root xylem and, consequently, from roots to leaves, ultimately impairing salt tolerance [80].


*Salix matsudana*


The analysis of molecular mechanisms regulating *SmEXPA13* expression in the salt tolerance of willow (*Salix matsudana* Koidz.) revealed the involvement of MYB and WRKY family transcription factors. Screening of a yeast library demonstrated that *SmMYB1R1-L* could bind the MYB element and regulate *SmEXPA13* expression. Accordingly, overexpression of *SmMYB1R1-L* in willow calli resulted in improved physiological parameters (higher fresh weight, lower REL, and reduced MDA content) compared to wild-type calli under salt stress. Furthermore, *SmMYB1R1* expression was higher in a salt-tolerant variety than in a salt-sensitive one under salt stress [81].

In addition to the salt tolerance mechanism in Salix, a further investigation into drought stress revealed the involvement of SmWRKY12 in modulating *SmEXPA13* expression through interaction with SmRAP2-7. Further yeast library screening also identified SmWRKY12 as a drought-induced protein capable of binding the *SmEXPA13* promoter and activating its expression. Further experiments revealed that SmWRKY12 can physically interact with SmRAP2-7, an AP2/ERF-type transcription factor, forming a dimer that enhances *SmEXPA13* transcription more effectively. Accordingly, overexpression of either the *SmWRKY12/SmRAP2-7* regulatory module or *SmEXPA13* markedly improved calli performance (higher fresh weight and lower REL) under drought stress compared to control plants [82]. These results demonstrate the crucial role of *SmEXPA13* in salt and drought stress tolerance and describe a novel regulatory mechanism in willow through the *SmMYB1R1-L* and *SmWRKY12/SmRAP2-7* modules.


*Glycine soja*


Expanding the scope to soybean, the expression of multiple expansin genes, such as *GsEXPA42* and *GsEXPB1* was significantly influenced by both salt and drought stress, demonstrating the species-specific roles of expansins in stress tolerance. Thus, among the 75 expansins identified in wild soybean (*Glycine soja* Siebold and Zucc.), salt stress upregulated the expression of *GsEXPA42*, *GsEXLA1*, *GsEXPB1*, *GsEXLB2*, and *GsEXLB14*, while it downregulated *GsEXPA3*, *GsEXPA36*, and *GsEXLB4*. Drought stress upregulated *GsEXPA38*, *GsEXPA42*, *GsEXPB7*, *GsEXLA1*, and *GsEXLB14*, while downregulating *GsEXPA36* expression. Overexpression of *GsEXPB1,* a root-specific gene, increased the number of hairy roots, root length, and root weight, resulting in improved salt stress tolerance [83].

Similarly, overexpression of *GsEXLB14*, another root-specific gene, increased root number, length, and weight under both salt and drought stress. Comparative transcriptome analysis of *GsEXLB14* overexpressing and control plants under salt stress revealed upregulation of genes encoding EXPB-type expansins, MYB transcription factors, auxin-responsive proteins, peroxidases, calcium/calmodulin-dependent protein kinases, H^+^-transporting ATPases, vacuolar membrane proton pumps, and anion channels. Under drought stress, genes encoding *peroxidases*, *calcium/calmodulin-dependent protein kinases*, *pathogenesis-related protein 1* (*PR1*), *expansin B/LB* family members, and *AP2*/*ERF* transcription factors were upregulated in transgenic plants compared to controls [84]. These findings highlight the close association between expansins and the expression of diverse genes involved in drought and salt tolerance, suggesting *GsEXPB1* and *GsEXLB14* as promising candidates for molecular breeding of stress-tolerant soybeans.


*Zea mays*


In maize (*Zea mays* L.), another important crop, the role of expansins in drought stress tolerance has also been explored. As a cross-pollinating crop, maize is particularly sensitive to water stress during the flowering stage, which can cause asynchronous development between the tassel and ear, thereby prolonging the anthesis-silking interval (ASI) and leading to yield loss. Notably, drought-susceptible haplotypes exhibited lower expression levels of *ZmEXPA5* compared to tolerant haplotypes. *ZmEXPA5* is primarily expressed in developing tissues, including the embryo, root, tassel, ear, and silk tissue. However, hormonal treatments (ABA, IAA, and zeatin) downregulated *ZmEXPA5* expression. Overexpression of *ZmEXPA5* in maize inbred line B104 reduced ASI and improved grain yield under both drought and well-watered conditions [85].


*Oryza sativa*


Similarly, in rice (*Oryza sativa* L.), the role of OsEXPA7 in modulating salt stress tolerance was underscored by its effect on cell wall modification and ion balance, with marked improvements in plant physiology under stress conditions. Under normal conditions, *OsEXPA7* is highly expressed in the shoot apical meristem, leaf sheath, shoot base, and root vasculature, particularly in the basal region and root hairs. *OsEXPA7* overexpressing lines exhibited enhanced salt stress tolerance, as indicated by greater root and shoot lengths, increased tiller numbers, reduced leaf damage and electrical conductivity, and higher chlorophyll content compared to WT plants. Morphological analysis revealed that *OsEXPA7* overexpression resulted in thicker and larger veins in leaf blades, an increased number of larger bundle sheaths and collenchyma cells in large veins, and longer metaxylem cells in primary roots compared to WT plants. Under salt stress, *OsEXPA7* overexpressing lines exhibited reduced Na^+^ and K^+^ accumulation and a lower Na^+^/K^+^ ratio in leaves and roots, while antioxidant enzyme activities (POD, APX, POX, and SOD) and proline accumulation were elevated, leading to reduced ROS production compared to WT plants. Transcriptome analysis revealed differential expression of genes involved in cation exchange, auxin signalling, cell-wall modification, and transcription regulation, including upregulation of *OsSOS1* (a sodium transporter), *OsARF23* (an auxin response factor), *OsWAK1* (a wall-associated receptor kinase), and *OsWRKY71* (a transcription factor). These findings suggest that *OsEXPA7* plays a vital role in enhancing salt stress tolerance by coordinating sodium transport, ROS scavenging, and cell-wall loosening [86].


*Triticum aestivum*


Like in rice, where OsEXPA7 modulates salt stress tolerance through cell wall modifications and ion balance, wheat expansins also contribute to stress adaptation by regulating cell wall properties under varying environmental conditions. Thus, salt stress differentially affects cell wall stiffness in the root elongation zone of salt-tolerant and salt-sensitive wheat cultivars by modulating expansin gene expression. Analysis of expansin gene expression under salt stress at different pH levels revealed that *TaEXPA5* expression was elevated at pH 6.0, contributing to cell wall loosening, while *TaEXPA8* was associated with cell wall loosening at pH 5.0. These findings indicate that different expansin genes mediate cell wall modifications in response to salt stress at varying pH levels [87].

As was shown in another study, plants overexpressing the wheat *TaEXPA2* gene exhibited enhanced drought stress tolerance, with increased fresh weight, plant height, chlorophyll content, net photosynthesis rate, and survival rate compared to WT plants. In contrast, *TaEXPA2* knockdown lines (via RNA interference) displayed a salt-sensitive phenotype. *TaEXPA2* overexpressing lines accumulated more proline and soluble sugars while exhibiting lower ROS, REL, and MDA levels compared to WT and RNAi lines under drought treatment. Antioxidant activities (SOD, APX, and CAT) and ROS scavenger gene expression (*TaMnSOD*, *TaAPX*, *TaCAT*, and *TaPOD*) were significantly higher in *TaEXPA2* overexpressing lines than in WT and RNAi plants under drought stress. A MYB family transcription factor, *TaMPS*, was identified as a direct activator of *TaEXPA2* expression. These findings suggest that *TaEXPA2* is a positive regulator of drought stress tolerance in wheat [88].

In this section, we highlight the functional characterisation of expansin mutants and transgenic plants, which highlights their significant contribution to enhancing drought and salt tolerance in crops such as soybean, maize, and rice. Genes like *GsEXPB1*, *ZmEXPA5*, and *OsEXPA7* have been shown to improve root growth and plant performance under stress conditions, thus indicating their vital role in abiotic stress resistance. For example, overexpression of *GsEXPB1* in soybean resulted in better root characteristics and salt tolerance, while *OsEXPA7* overexpression in rice led to enhanced salt stress resilience. Furthermore, the positive regulation of expansin genes by transcription factors, including MYB and AP2/ERF families, suggests an intricate network of gene interactions that mediate stress responses. These findings provide strong evidence that manipulating expansin expression can lead to improved crop resilience, supporting the potential for expansins as key targets in agricultural biotechnology. The functional roles of expansins across different species further affirm their broad application for improving stress tolerance in various crops. Taken together, these findings across different plant species highlight the multifaceted roles of expansin genes in regulating both abiotic and biotic stress responses, with varying outcomes depending on the specific expansin and species involved.

## 5. Integrative Analysis and Future Perspectives

The analysis of expansin function in various plant species highlights their crucial role in salt and drought stress tolerance through multiple interconnected mechanisms. Despite differences in genetic backgrounds and environmental adaptations, several common pathways emerge across species. Expansins primarily modulate cell wall loosening, facilitating root elongation and maintaining cellular integrity under stress conditions. In several cases, they also interact with transcription factors such as MYB, WRKY, and AP2/ERF, forming regulatory networks that coordinate stress responses. Additionally, expansins contribute to ion homeostasis, water retention, and hormonal signalling, enabling plants to withstand adverse environmental conditions.

A recurrent theme in stress adaptation involves the regulation of antioxidant defence systems and ROS scavenging. Many expansin-overexpressing lines exhibited enhanced activities of antioxidant enzymes, including SOD, POD, APX, and CAT. These enzymes mitigate oxidative damage by neutralising excess ROS, thereby reducing lipid peroxidation and maintaining membrane stability under stress conditions. For example, overexpression of *TaEXPA2* in wheat and *OsEXPA7* in rice was associated with increased SOD and APX activity, leading to enhanced oxidative stress tolerance. The observed increase in ROS scavenging capacity suggests that expansins may indirectly modulate redox homeostasis, either through interactions with ROS-related transcription factors or by influencing cell wall-associated oxidative reactions.

In addition to antioxidant defence, expansins have been implicated in the accumulation of osmoprotectants, including soluble sugars and proline, which play a critical role in stress tolerance. Enhanced levels of proline and soluble sugars were observed in multiple expansin-overexpressing transgenic lines, contributing to osmotic adjustment, water retention, and membrane protection. For instance, overexpression of *NtEXPA4* in *Nicotiana tabacum* and *BrEXLB1* in *Brassica rapa* resulted in significant proline accumulation, leading to improved drought tolerance. These molecules act as compatible solutes, stabilising proteins and cellular structures under dehydration stress while also serving as energy reserves. The correlation between expansin expression and osmoprotectant accumulation suggests a broader role for these proteins in maintaining cellular homeostasis beyond their primary function in cell wall remodelling.

Phytohormones play a pivotal role in expansin-mediated stress responses, influencing growth, development, and adaptation to environmental stressors. Expansins have been shown to interact with ABA, ethylene, auxins, and gibberellins, modulating downstream signalling pathways that regulate stomatal closure, root architecture, and stress-responsive gene expression. For example, ABA-induced expression of *TaEXPA2* in wheat was linked to increased drought resistance, while auxin-associated expression of *OsEXPA7* in rice facilitated adaptive growth responses under salt stress. Similarly, ethylene-responsive expansins, such as *GsEXLB14* in soybean, play a role in maintaining cellular expansion under water deficit conditions. The interplay between expansins and phytohormones underscores their complex role in integrating hormonal signalling with mechanical stress adaptation, warranting further investigation into these regulatory networks.

Despite the progress in understanding expansin-mediated stress tolerance, several limitations remain. Many studies focus on single-gene overexpression or knockdown approaches, which may not fully capture the complexity of expansin interactions within broader signalling networks. Additionally, the downstream molecular targets of expansins remain largely unexplored, particularly regarding their role in transcriptional regulation, hormone crosstalk, and secondary metabolite synthesis. Future research should aim to unravel these connections through integrative omics approaches, including transcriptomics, proteomics, and metabolomics, to provide a more comprehensive understanding of expansin function.

Furthermore, most studies have been conducted under controlled laboratory conditions, limiting their applicability to field environments where multiple stress factors interact. Expanding research to include field trials and multi-environment analyses would provide valuable insights into the practical utility of expansins for crop improvement. Additionally, genome editing technologies such as CRISPR/Cas could be leveraged to precisely manipulate expansin genes and validate their roles in stress adaptation across diverse plant species. Another promising avenue involves exploring epigenetic regulation of expansin genes under stress conditions, which could reveal new targets for improving plant resilience. Moreover, engineering expansins in combination with other stress-responsive genes, such as aquaporins or osmoprotectant-related enzymes, could lead to synergistic improvements in stress tolerance.

Taken together, these findings underscore the importance of expansins as key modulators of abiotic stress responses, with promising implications for crop breeding. By integrating knowledge from different species and leveraging advanced molecular techniques, future research can further refine expansin-based strategies to enhance plant resilience against environmental challenges.

## 6. Conclusions

This manuscript provides a comprehensive analysis of expansins and their diverse roles in plant responses to salt and drought stress. Across different species, expansins have been shown to modulate cell wall loosening, regulate transcriptional networks, influence ion and osmotic homeostasis, and interact with key phytohormonal pathways to enhance stress resilience.

In *Nicotiana tabacum*, overexpression of *NtEXPA4* and *NtEXPA11* promoted root elongation and ion balance under salt and drought stress, while in *Brassica rapa*, *BrEXLB1* influenced root structure and stress adaptation through interactions with phytohormones. In *Oryza sativa*, *OsEXPA7* was linked to antioxidants, proline, cation exchange, auxin signalling, and cell-wall modification pathways under salt stress conditions, and *OsEXPA7* association with the auxin signalling pathway further reinforced the role of expansins in integrating mechanical and hormonal responses. Meanwhile, *Populus trichocarpa PtEXPA6* was identified as a negative regulator of salt stress tolerance, highlighting the functional diversity of expansins across species.

Expansins also contribute to ROS homeostasis, as seen in transgenic lines, with enhanced activity of SOD, POD, APX, and CAT, mitigating oxidative damage. Furthermore, the increased accumulation of soluble sugars and proline in expansin-overexpressing plants suggests their involvement in osmotic adjustment mechanisms. The crosstalk between expansins and phytohormones, particularly ABA, auxins, and ethylene, underscores their role in coordinating growth and stress responses.

Taken together, these findings highlight expansins as the key modulators of plant adaptation to environmental stress, with species-specific roles that shape their functional outcomes. Further research is needed to elucidate the precise molecular mechanisms governing expansin activity, their interactions with stress-responsive pathways, and their potential applications in improving crop resilience.

## Figures and Tables

**Figure 1 plants-14-01327-f001:**
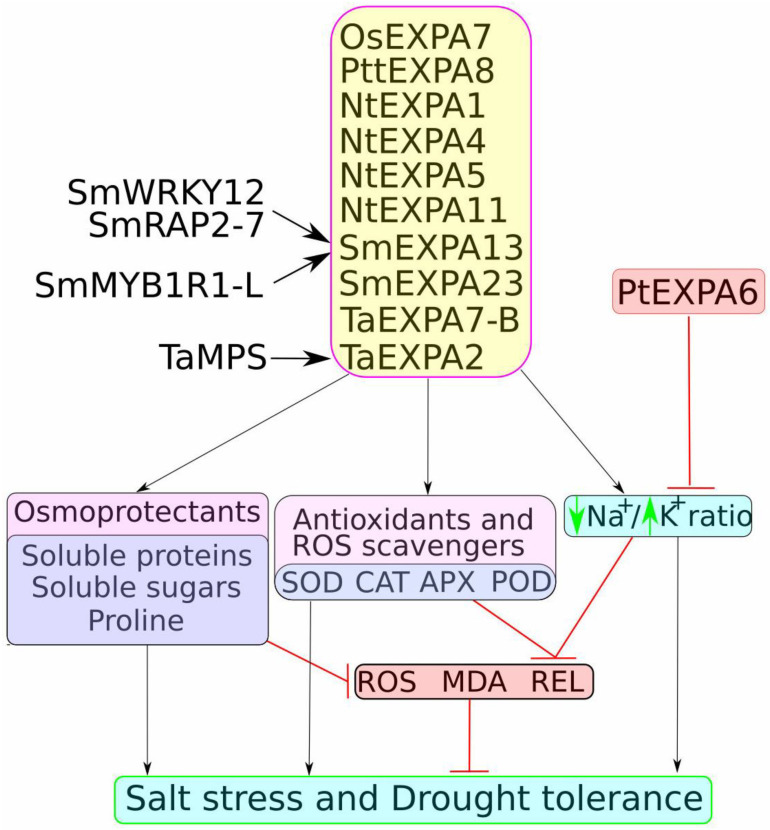
Molecular pathways associated with expansin-mediated enhancement of plant tolerance to salt and drought stress. Expansins positively regulate the production and accumulation of osmoprotectants (such as proline, soluble sugars, and proteins), the expression and activity of antioxidants and reactive oxygen species (ROS) scavengers (such as SOD, POD, CAT, and APX), and contribute to the maintenance of the Na^+^/K^+^ ratio. SmMYB1R1-L, SmWRKY12, and SmRAP2-7 have been identified as transcription factors regulating *SmEXPA13* expression, while TaMPS has been identified as a positive regulator of *TaEXPA2*; however, the upstream regulators of most expansins remain unknown. *PtEXPA6* has been characterised as a negative regulator of salt and drought tolerance. Black arrows indicate positive regulation, while blunt red lines indicate negative regulation.

**Figure 2 plants-14-01327-f002:**
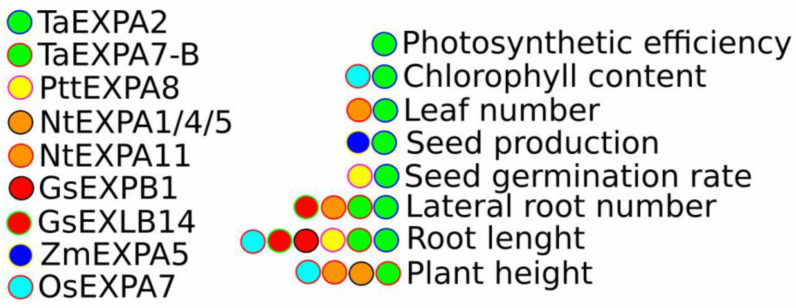
Phenotypic characteristics associated with ectopic or native expression of various expansins. Each gene is responsible for a specific phenotypic feature (coloured circles connect genes and corresponding phenotypes).

**Table 1 plants-14-01327-t001:** Expansin genes associated with salt and drought stress tolerance in different plant species.

Plant Species	Total Expansisns in Genome *	Effects of Stress and Hormones on Expansin Gene(s)	References
*Nicotiana tabacum*	58	Salt, drought, heat, and cold stresses and ABA, CK, Auxin, and gibberellins upregulated *NtEXPA1,4*,5 expression	[65]
potato (*Solanum tuberosum* L.)	38	IAA, GA_3_, and BAP upregulated *StEXPA7/18;* ABA and GA_3_ upregulated all *StEXLB* genes; NaCl and mannitol upregulated *StEXPA8/19* and *StEXPB2,* and downregulated *StEXPA4* and *StEXLB4*; Any stress or hormonal treatment affected *StEXPA1/21/23/24*, and *StEXPB5*	[66]
wheat (*Triticum* *aestivum* L.)	275	PEG treatment in leaves upregulated *TaEXPA3/9-A*, *TaEXPB2/4/7/9/10-A*; salt treatment upregulated *TaEXPA3/9-A*, *TaEXPB2/4/10-A*; PEG treatment in roots upregulated *TaEXPA3-8-A*, *TaEXPB1/7/8/10-A*, and *TaEXPB1-B*; Salt stress upregulated *TaEXPA3/5-8-A*, *TaEXPA12-A*, *TaEXPB2/4/7/10-A*	[67]
barley (*Hordeum* *vulgare* L.)	45	Drought stress upregulated *HvEXPB5/6* in roots	[68]

* The number of known genes was accessed from the InterPro database [69].

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
