# Peer review of "Expansins in Salt and Drought Stress Adaptation: From Genome-Wide Identification to Functional Characterisation in Crops"

_plants, 2025, doi:10.3390/plants14091327_

Round 1
Reviewer 1 Report
Comments and Suggestions for Authors
Comments to the Author
The roles of expansin members in salt and drought stress is novelty and significant. The content in text is some sufficient, however the version might be revised for more logical and the organization paragraphs and sentences might be adjusted. The similar discriptions could be merged.
- though the researches about the expansin in the origin paper were investigated by genome-wide identification, however, in this paper, how to understand the explanation about genome-wide identification?
- the expansins from different plant species were independent, if there are phylogenic relationship between TaEXPA2, NtEXPA1, ZmEXPA5, and so on, why did these expansins not others response to salt and drought stresses.
- part 3. Functional Analysis of Expansins in Salt and Drought Stress Tolerance, more focused and some relationship?
Therefore, I recommend the authors to revise the manuscript carefully and resubmit the improved version.
Author Response
Dear Editor and Reviewers,
We greatly appreciate your critical evaluation of our manuscript and helpful comments. Our reply to your comments would be provided point by point, where “A” stands for “Authors”, and “L” for “Lines”, where changes have been implemented.
____________________________________________________________________________
The roles of expansin members in salt and drought stress is novelty and significant. The content in text is some sufficient, however the version might be revised for more logical and the organization paragraphs and sentences might be adjusted. The similar discriptions could be merged.
A: Thank you for recognizing the novelty and significance of our work. We have carefully revised the manuscript for improved clarity, logical flow, and organization. Paragraphs and sentences have been restructured where necessary, and redundant descriptions have been merged to enhance readability. If we have misunderstood any part of your comment, we would appreciate further clarification—particularly regarding expressions such as “text is some sufficient” and “organization paragraphs,” which are a bit unclear to us.
- though the researches about the expansin in the origin paper were investigated by genome-wide identification, however, in this paper, how to understand the explanation about genome-wide identification?
A: Thank you for your observation. If this comment refers to Section 2 of our manuscript, it summarizes expansin studies identified through genome-wide analyses in multiple plant species (e.g., tobacco, potato, wheat, and barley). These references help provide context for our study. If this is not what you meant, we kindly ask for a rephrased version of your comment so that we can address it more precisely
- the expansins from different plant species were independent, if there are phylogenic relationship between TaEXPA2, NtEXPA1, ZmEXPA5, and so on, why did these expansins not others response to salt and drought stresses.
A: Thank you for this insightful question. Indeed, it raises an important point. However, the cited studies do not provide phylogenetic analyses or detailed molecular explanations of why specific expansins (such as TaEXPA2, NtEXPA1, ZmEXPA5) respond to salt and drought stress while others do not. As noted in Section 4 of our manuscript, many of these works are descriptive and focus primarily on phenotypic changes in mutants, without delving into molecular mechanisms or evolutionary relationships. We agree this is an area where further research is needed
- part 3. Functional Analysis of Expansins in Salt and Drought Stress Tolerance, more focused and some relationship?
- A: If this question refers to the relationship between different expansins discussed in Part 3, we acknowledge that such comparative or functional links would be valuable. Unfortunately, the currently available data in the cited literature do not provide direct evidence of such relationships. Consequently, we are limited in our ability to elaborate further. We hope future research will offer more insight into this aspect.
Therefore, I recommend the authors to revise the manuscript carefully and resubmit the improved version.
A: Thank you for your valuable comments.

Reviewer 2 Report
Comments and Suggestions for Authors
Major concerns
- There should be an introduction part at the beginning of the manuscript.
- It is better to list genes responsive to different stresses in a table in part 2.
- Figure 2 should be rephrased, which is not visual enough.
- It is not thorough enough to list results from several crops in part 3.2. The authors should try to find connections at evolutionary levels. Are these reported genes homologous or not? Are there similarities in protein sequences of these genes? Are there connections between protein structures and functions? It is much more important to get new findings rather than description.
- The authors should be more cautious in conclusion part. Expansins have connections with abiotic stresses. But it is hard to say that expansin is the result or the reason. If it is the reason, then how it changes after abiotic stresses? Besides, phytohormones should be the reason, since their concentrations are easily affected when cell dehydration after abiotic stresses, which leads response of receptors. They should be more important than expansin in the network.
Author Response
Dear Editor and Reviewers,
We greatly appreciate your critical evaluation of our manuscript and helpful comments. Our reply to your comments would be provided point by point, where “A” stands for “Authors”, and “L” for “Lines”, where changes have been implemented.
____________________________________________________________________________
- There should be an introduction part at the beginning of the manuscript.
A: Thank you for your suggestion. An Introduction section has now been included at the beginning of the manuscript to provide background information and better frame the objectives of this review .
- It is better to list genes responsive to different stresses in a table in part 2.
A: We appreciate this helpful suggestion. To improve clarity and facilitate understanding, we have added Table 1, which summarizes expansin genes reported to respond to various abiotic stresses.
- Figure 2 should be rephrased, which is not visual enough.
A: Thank you for pointing this out. We have expanded the caption for Figure 2 to better explain the visual elements and to improve clarity. If further graphical adjustments are needed, we would be grateful for more specific suggestions.
- It is not thorough enough to list results from several crops in part 3.2. The authors should try to find connections at evolutionary levels. Are these reported genes homologous or not? Are there similarities in protein sequences of these genes? Are there connections between protein structures and functions? It is much more important to get new findings rather than description.
A: Thank you for this thoughtful comment. We fully agree that such comparative analyses would add substantial value. However, as this is a review article, our objective is to synthesize findings from previously published research rather than to present new experimental data. To our knowledge, there are currently no comprehensive studies that clearly establish evolutionary or structural relationships between expansin proteins and their roles in drought/salt tolerance. We have mentioned this gap in knowledge in Section 4 of the manuscript, emphasizing the need for future studies that explore these connections more deeply.
- The authors should be more cautious in conclusion part. Expansins have connections with abiotic stresses. But it is hard to say that expansin is the result or the reason. If it is the reason, then how it changes after abiotic stresses? Besides, phytohormones should be the reason, since their concentrations are easily affected when cell dehydration after abiotic stresses, which leads response of receptors. They should be more important than expansin in the network.
A: Thank you for this important comment. We agree that the conclusion should be stated more cautiously, and we have revised it accordingly. We now emphasize that expansins are involved in the response to abiotic stress rather than being a direct cause. Additionally, we recognize the significant role of phytohormones in abiotic stress responses and have added a note on their upstream influence, particularly in relation to cellular dehydration and receptor signaling. However, we would appreciate a clarification of certain parts of your comment—particularly the phrase "expansins are the result or the reason"—to ensure we fully address your concern.

Reviewer 3 Report
Comments and Suggestions for Authors
The review article "Expansins in Salt and Drought Stress Adaptation: From Genome-Wide Identification to Functional Characterisation in Crops" provides a comprehensive analysis of the critical role of expansins—cell wall-modifying proteins—in enhancing plant resilience to abiotic stresses. By examining genome-wide identification and functional characterizations across species, the authors highlight expansins' involvement in transcriptional regulation, ion and osmotic homeostasis, phytohormone signaling, and reactive oxygen species mitigation. Their findings highlight expansins' potential in crop improvement strategies aimed at countering salt and drought stress.
In this manuscript, the abstract (lines 17–23) primarily reiterates the results of specific studies. The authors are encouraged to adopt a different approach for reviewing these works to enable a synthesis of knowledge. This recommendation applies to the manuscript as a whole. Instead of summarizing previous studies and findings, the focus should shift toward integrating and synthesizing knowledge.
The review article lacks an Introduction section. Although the background information provided in Section 1 resembles an introduction, a well-structured Introduction section would be a valuable addition. This section could include background context, current deficiencies in compiling reviews on this topic, and the motivation behind conducting this review. This omission is especially apparent when the authors transition directly to their discussion in lines 97–107 without addressing existing gaps or the rationale for this review.
Stomatal movement is a crucial aspect of drought tolerance, and plant cell walls play an essential role in stomatal development and dynamics. The authors should consult and cite the following article: doi: 10.1016/j.ncrops.2024.100021. Additionally, it would be beneficial to establish connections between expansins, cell walls, stomata, and drought tolerance.
In lines 47–56, the domain structures described should be accompanied by visual presentations using graphics. A tabular presentation would also help readers better understand the number and types of domains, as well as the number of genes present across different species.
Similarly, in section 2, “Genome-Wide Identification and Expression Analysis Under Stress Conditions in Model and Crop Species,” the text is overly exhaustive. The authors should consider presenting this information in tables to make the content more accessible and easier for readers to follow.
The caption for Figure 2 is too brief. The meaning of the different colored circles is unclear. Expand the caption to make the figure self-explanatory.
In section 3, “Functional Analysis of Expansins in Salt and Drought Stress Tolerance,” the inclusion of more illustrations incorporating graphics would enhance the readers’ understanding. A figure showcasing the involvement of different hormones and their mechanisms would also be helpful.
Although the authors refer to breeding in several places (lines 35, 398, and 540), the discussion remains inadequate. The authors should elaborate on how the synthesized knowledge can be applied in breeding programs aimed at improving drought and salt tolerance. Furthermore, the mechanism of stress tolerance should be explored in greater detail. The authors are suggested to consult and cite the following article: doi: 10.1016/j.ncrops.2024.100029.
The review focuses on the role of expansins in drought and salt stress tolerance. However, these two stresses are discussed together. The authors should include a section that distinguishes and compares plant responses to drought and salt stress, particularly in relation to expansins.
Lastly, the manuscript contains several instances of repetitive information. The authors should remove redundant text and messages to streamline the content.
Author Response
Dear Editor and Reviewers,
We greatly appreciate your critical evaluation of our manuscript and helpful comments. Our reply to your comments would be provided point by point, where “A” stands for “Authors”, and “L” for “Lines”, where changes have been implemented.
____________________________________________________________________________
The review article "Expansins in Salt and Drought Stress Adaptation: From Genome-Wide Identification to Functional Characterisation in Crops" provides a comprehensive analysis of the critical role of expansins—cell wall-modifying proteins—in enhancing plant resilience to abiotic stresses. By examining genome-wide identification and functional characterizations across species, the authors highlight expansins' involvement in transcriptional regulation, ion and osmotic homeostasis, phytohormone signaling, and reactive oxygen species mitigation. Their findings highlight expansins' potential in crop improvement strategies aimed at countering salt and drought stress.
In this manuscript, the abstract (lines 17–23) primarily reiterates the results of specific studies. The authors are encouraged to adopt a different approach for reviewing these works to enable a synthesis of knowledge. This recommendation applies to the manuscript as a whole. Instead of summarizing previous studies and findings, the focus should shift toward integrating and synthesizing knowledge.
A: Thank you for this insightful suggestion. As a review article, the primary aim of the manuscript is to summarize, compare, and discuss findings from previously published studies. While we acknowledge the value of a synthesized perspective, our discussion is inherently constrained by the descriptive nature of available research on expansins. We have made efforts to highlight patterns and knowledge gaps across studies, but the generation of novel findings or deeper synthesis typically requires new experimental work, which falls within the scope of original research articles.
The review article lacks an Introduction section. Although the background information provided in Section 1 resembles an introduction, a well-structured Introduction section would be a valuable addition. This section could include background context, current deficiencies in compiling reviews on this topic, and the motivation behind conducting this review. This omission is especially apparent when the authors transition directly to their discussion in lines 97–107 without addressing existing gaps or the rationale for this review.
A: Thank you for pointing this out. A dedicated Introduction section has now been added. It provides relevant background, outlines the current gaps in expansin-focused reviews, and clarifies the motivation and scope of the manuscript.
Stomatal movement is a crucial aspect of drought tolerance, and plant cell walls play an essential role in stomatal development and dynamics. The authors should consult and cite the following article: doi: 10.1016/j.ncrops.2024.100021. Additionally, it would be beneficial to establish connections between expansins, cell walls, stomata, and drought tolerance.
A: We appreciate the recommendation. While stomatal movement is indeed relevant to drought responses, the suggested article does not address expansins or their role in stomatal regulation. To date, no direct mechanistic link has been established between expansins and stomatal dynamics under abiotic stress. We believe including unrelated references may dilute the manuscript's focus. However, should future studies explore such connections, we agree that their inclusion in a review would be highly valuable.
In lines 47–56, the domain structures described should be accompanied by visual presentations using graphics. A tabular presentation would also help readers better understand the number and types of domains, as well as the number of genes present across different species.
A: Thank you for the suggestion. Expansins contain two highly conserved domains (D1 and D2) connected by a linker. Due to this simple and well-established structure, we believe that adding a separate figure or table may not significantly enhance understanding. Nevertheless, we have included the number of expansin genes across selected species in Table 1 for improved clarity.
Similarly, in section 2, “Genome-Wide Identification and Expression Analysis Under Stress Conditions in Model and Crop Species,” the text is overly exhaustive. The authors should consider presenting this information in tables to make the content more accessible and easier for readers to follow.
A: Table 1 was added.
The caption for Figure 2 is too brief. The meaning of the different colored circles is unclear. Expand the caption to make the figure self-explanatory.
A: The caption for Figure 2 was modified.
In section 3, “Functional Analysis of Expansins in Salt and Drought Stress Tolerance,” the inclusion of more illustrations incorporating graphics would enhance the readers’ understanding. A figure showcasing the involvement of different hormones and their mechanisms would also be helpful.
A: This is an excellent point. However, the currently available literature provides limited data on hormonal regulation of expansins under drought and salt stress conditions. While a few studies mention the involvement of ABA, GA, IAA, and CKs, these references do not offer enough detail to support an accurate or comprehensive figure. As highlighted in Section 4, this lack of molecular insight is a limitation in current expansin-related research.
Although the authors refer to breeding in several places (lines 35, 398, and 540), the discussion remains inadequate. The authors should elaborate on how the synthesized knowledge can be applied in breeding programs aimed at improving drought and salt tolerance. Furthermore, the mechanism of stress tolerance should be explored in greater detail. The authors are suggested to consult and cite the following article: doi: 10.1016/j.ncrops.2024.100029.
A: Thank you for this suggestion. We fully agree that translating current knowledge into breeding strategies is essential. However, the cited article does not discuss expansins and appears to fall outside the core scope of this review. Furthermore, the current literature on expansins remains largely descriptive, with limited insights into mechanisms relevant for breeding applications. We have addressed these limitations in Section 4, emphasizing the need for more integrative and mechanistic studies that could inform breeding programs in the future.
The review focuses on the role of expansins in drought and salt stress tolerance. However, these two stresses are discussed together. The authors should include a section that distinguishes and compares plant responses to drought and salt stress, particularly in relation to expansins.
A: Thank you for this perspective. We have chosen to discuss salt and drought stress together because many expansin genes are responsive to both stress types, often within the same study. Separating them could lead to redundancy and fragmentation. Nonetheless, we have taken care to indicate stress-specific roles where such distinctions are clear in the literature.
Lastly, the manuscript contains several instances of repetitive information. The authors should remove redundant text and messages to streamline the content.
A: Thank you for bringing this to our attention. The manuscript has been carefully revised to eliminate redundant content and improve conciseness throughout.

Round 2
Reviewer 1 Report
Comments and Suggestions for Authors
accept
Reviewer 2 Report
Comments and Suggestions for Authors
The authors have addressed most of the comments. The manuscript could be accepted after grammar checking.
Reviewer 3 Report
Comments and Suggestions for Authors
The revised article can be accepted in its present form